# Modification of a Tumor-Targeting Bacteriophage for Potential Diagnostic Applications

**DOI:** 10.3390/molecules26216564

**Published:** 2021-10-29

**Authors:** Maya Alexandrovna Dymova, Yaroslav Alexandrovich Utkin, Maria Denisovna Dmitrieva, Elena Vladimirovna Kuligina, Vladimir Alexandrovich Richter

**Affiliations:** Institute of Chemical Biology and Fundamental Medicine, Siberian Branch of the Russian Academy of Sciences, 630090 Novosibirsk, Russia; utkinyaroslav99@gmail.com (Y.A.U.); imaria819@gmail.com (M.D.D.); kuligina@niboch.nsc.ru (E.V.K.); richter@niboch.nsc.ru (V.A.R.)

**Keywords:** MDA-MB 231 cells, tumor-targeting bacteriophages, chemical modification of bacteriophages, FAM–NHS, theranostic drugs

## Abstract

Background: Tumor-targeting bacteriophages can be used as a versatile new platform for the delivery of diagnostic imaging agents and therapeutic cargo. This became possible due to the development of viral capsid modification method. Earlier in our laboratory and using phage display technology, phages to malignant breast cancer cells MDA-MB 231 were obtained. The goal of this study was the optimization of phage modification and the assessment of the effect of the latter on the efficiency of phage particle penetration into MDA-MB 231 cells. Methods: In this work, we used several methods, such as chemical phage modification using FAM-NHS ester, spectrophotometry, phage amplification, sequencing, phage titration, flow cytometry, and confocal microscopy. Results: We performed chemical phage modification using different concentrations of FAM-NHS dye (0.5 mM, 1 mM, 2 mM, 4 mM, 8 mM). It was shown that with an increase of the modification degree, the phage titer decreases. The maximum modification coefficient of the phage envelope with the FAM–NHS dye was observed with 4 mM modifying agent and had approximately 804,2 FAM molecules per phage. Through the immunofluorescence staining and flow cytometry methods, it was shown that the modified bacteriophage retains the ability to internalize into MDA-MB-231 cells. The estimation of the number of phages that could have penetrated into one tumor cell was conducted. Conclusions: Optimizing the conditions for phage modification can be an effective strategy for producing tumor-targeting diagnostic and therapeutic agents, i.e., theranostic drugs.

## 1. Introduction

Cancer is a leading cause of death, and the cancer incidence and cancer death rate are rising steadily [1]. Standard cancer treatment methods, including surgery, chemotherapy, and radiation are sometimes ineffective and are limited by the lack of a high selectivity of action against cancer cells, which leads to significant damage to the patient’s body, even in cases of complete remission [2].

Currently, bacteriophages are a promising tool in the field of cancer therapy. The filamentous bacteriophage, due to its ability to display or bind various molecules on the surface of the capsid and an innate capability to penetrate and traverse tissues and barriers, can be effectively used to deliver drug delivery vehicles for cancer treatment [3]. The phage display technique makes it possible to find bacteriophages displaying tumor-targeting peptides on their capsid shell [4]. The mechanism of internalization, intracellular transport, and stability of tumor-targeting filamentous phage M13 inside a eukaryotic cell was described earlier [5]. The internalization of phages mainly occurs because of the endocytic mechanism through specific receptors, which allows the delivery and accumulation of drugs directly inside the cell.

Filamentous bacteriophages are used more often in the phage display technique. Their main advantage over other phages is their relatively high productivity—their titer is about 100 times higher than that of any other known phage [6]. Additionally, filamentous phages have greater and prolonged penetration compared to spherical phages [7]. Filamentous phages are stable over a wide pH range (3–11) and can withstand heating to 80 °C, which allows them to perform a wide range of phage modifications [8]. A widely used strategy in filamentous phage chemical modification is primarily based on the nonspecific modification of the free amino groups of the phage coat proteins, which are the most reactive [9,10,11,12,13,14]. 

One of the most commonly used modification strategies is the use of *N*-hydroxysuccinimide esters (NHS), which are described in Figure 1 [15,16]. 

To understand how protein modification occurs using NHS esters, an empirical kinetic model has been developed to predict the modification level of filamentous phages over a wide range of reaction conditions. It was shown that 50% of the maximum ability of the phage to bind to streptavidin is achieved with the attachment of approximately 0.03 biotin per one subunit of a major coat protein (pVIII) of the M13 phage [17]. In subsequent studies, using UV-visible spectroscopy, it was found that up to 1600 TMR-molecules (tetramethylrhodamine) can be attached to one M13 bacteriophage [18]. The main problem is that when a phage particle carries more than 1600 TMR molecules, fluorescence quenching occurs due to the formation of non-fluorescent H-dimers. In addition, the study showed that some of the six amines presented in the major coat protein pVIII (*N*-terminal Ala-1 and Lys 8, 40, 43, 44 and 48) were more reactive than others. When the modifying agent TMR-NHS is at low concentrations, it is mainly Ala-1 that enters the reaction, and at high concentrations, both Ala-1 and Lys-8 are modified. This reaction optimally proceeds at a pH between 8.0–8.5 and leads to the formation of a stable amide bond after the release of the NHS group [19]. 

Despite the reliability of this modification strategy, this approach can lead to undesirable hypermodification and, as a consequence, a decrease in the penetrating ability of the phages. Therefore, it is necessary to both select conditions for the modification of tumor-targeting phages and to investigate the penetration degree of such phages into the tumor cell. In this work, using the chemical modification of the free amino groups of the phage coat proteins (pVIII) with FAM-NHS ester, we estimated the phage modification coefficient with the FAM fluorophore (FAM/phage) and qualitatively showed the binding and penetration efficiency of modified phages as well as the maximum amount bacteriophages that are capable of internalizing into tumor cells. In further studies, the modified bacteriophage can be conjugated with an oncolytic drug, which will make it possible to use this modified phage as a theranostic drug. 

## 2. Results

### 2.1. Obtaining Modified Bacteriophages M13-FAM

The amount of bacteriophages M13 required for chemical modification was amplified in cultures of *E. coli* strain ER2738. The titer of the obtained phages was determined using the soft-agar overlay technique, and it was 10^13^ PFU/mL.

Sanger sequencing was performed to ensure that we amplified a bacteriophage M13 displaying a tumor-targeting peptide (YTYDPWLIFPAN). This was confirmed by the presence of the insert corresponding to the previously selected peptide (Figure 2).

Further, FAM-modified bacteriophages were obtained and purified from the unbound dye. The purification was conducted by a three-fold precipitation with PEG/NaCl at 4 °C.

The data on the viability of bacteriophages modified at different FAM-NHS dye concentrations (0.5 mM, 1 mM, 2 mM, 4 mM, 8 mM) were obtained using the soft-agar overlay technique. When the modifying agent concentration increased, a decrease in the titer of the modified M13 phage was observed (Table 1); during the modification process, the titer decreased by 1.5 times (M13 (control)) compared to the titer, which was on the start of the experiment (M13 (initial)). 

At the same time, the DMSO present in a solution also affects the titer, and it decreases by about 15 times as a result of this compared to the titer of the original phage (M13 (initial)). Thus, it can be concluded that the concentration of both DMSO and the FAM fluorophore affects the titer of the bacteriophages. 

### 2.2. Spectrophotometric Analysis of Experimental Samples

The spectra of the experimental samples had an absorption maximum at 494 nm, which corresponded to the literature values of the maximum absorption of FAM-NHS dye (Figure 3). The peak at 269 nm corresponds to the absorption of M13 bacteriophages [20]. The spectra of the control samples, in contrast to the experimental ones, did not have an absorption peak at 494 nm, which indicates the modification of the bacteriophage shell with the FAM-NHS dye. 

At the same time, we evaluated the absorption spectra of all of the washes after the last precipitation. We did not see any absorption at 494 nm, which indicated a complete washing of the unbound FAM molecules (data not shown). 

### 2.3. Evaluation of the Loading of FAM Fluorophore on M13 Bacteriophage 

UV-vis spectrophotometry was used to determine the modification coefficient of the M13 bacteriophage coat protein by the FAM fluorophore based on the molar absorption of the dye and of the virus according to Formulas (1)–(3). The modification coefficient was determined. The results are shown in Figure 4. 

As seen from Figure 3, when the concentration of the fluorophore FAM-NHS in the reaction mixture consisted of 4 mM, the modification coefficient of the phages (FAM/phage) was 804.2 dye molecules attached to one phage particle. When the modifying agent was at a high concentration (8 mM FAM-NHS), the degree of modification decreased to 726.2 dye molecules/phage. There is a possibility that is phenomenon is caused by the formation of FAM fluorophore dimers, leading to a change in the optical properties. A similar phenomenon was shown for TAMRA dye by Li K. et al. [18].

### 2.4. Analysis of the Binding Efficiency of the Modified Bacteriophage to Tumor Cells

The analysis of the binding efficiency of the modified bacteriophage to tumor cells MDA-MB 231 was performed using flow cytometry (Figure 5). The proportion of stained cells increased with increasing fluorophore concentration, with the maximum being observed for the sample “4 mM Fam-NHS + phages” (98.4%); it is worth noting that the histogram corresponding the sample “4 mM Fam-NHS + phages” was higher and narrower than that for the sample “8 mM Fam-NHS + phages”, which could point to the more specific binding of these phages to the tumor cells MDA-MB 231. 

### 2.5. Analysis of the Modified Bacteriophage Penetrating Ability into Tumor Cells

The ability of the M13-FAM bacteriophage to penetrate into tumor cells was investigated by immunofluorescence staining and confocal microscopy. MDA-MB-231 cells were incubated with M13-FAM bacteriophages modified at various concentrations of the fluorescent dye (0.5 mM, 1 mM, 2 mM, 4 mM, 8 mM). It was shown that these bacteriophages also retained the ability to penetrate into tumor cells (Figure 6) and that the degree of modification was directly proportional to the signal intensity in the green channel corresponding to the FAM dye. 

The change in the intensity of the red channel could be associated with competition between the fluorophore and antibodies for binding sites on the 2700 copies of the major coat protein pVIII. This may indicate that the modification takes place at the amino groups of the pVIII proteins. It can also be seen from Figure 5 that a certain amount of the FAM fluorophore (green channel) that conjugated to the M13 phage (red channel) is located in the intercellular space. In order to determine the nature of this phenomenon, an experiment was conducted in which the M13-FAM bacteriophage and M13 phage were incubated on 8-well slides in the absence of cells, and confocal microscopy was performed. The M13-FAM bacteriophage was better adsorbed on the glass surface than it was on M13 (control), which suggests that the FAM modification of the phage shell increases the tropism of the modified bacteriophage to the glass surface (data not shown); in the future, this must be borne in mind when choosing a fluorophore. 

The projection of a three-dimensional image (Figure 7) shows that the M13-FAM bacteriophages are located in the cytoplasm of tumor cells. This is indicated by the presence of a signal in the green channel in the immediate vicinity of the nucleus and the outlines of bright green vesicles, which contain a bacteriophage, in the cytoplasm.

### 2.6. Evaluation of Amount of the Bacteriophages Penetrated into Tumor Cells

To assess the number of phages that entered tumor cells, we incubated a fixed number of cells with a different number of unmodified and 4 mM FAM-modified phages: in the experiment, the ratio of phages to cells in excess (10, 100, 1000, 10,000, 100,000 times) were studied. The incubation conditions for the cells with the phages as well as further washes were performed according to the protocols described for flow cytometry. After incubation, we washed the sample three times to remove any unbound phages. The following ratio of the number of phages to the number of cells was obtained (Figure 8): 

There was an excess of phages in relation to cells by a factor of 100,000. The number of penetrated unmodified phage particles was 476 per 1 tumor cell. At the same time, the number of penetrated modified phage particles was dramatically lower (96 phages/cell). We assume that a tumor-targeting peptide can be modified with a fluorescent dye, which leads to a decrease in the affinity of the modified phages for tumor cells.

## 3. Discussion

Bacteriophages could be used as vehicles for the development of theranostics or imaging agents. The main advantages of these molecules are their size, high solubility, and multivalency. The latter is important because both the payload and signaling motifs can be conjugated with the shell of the virus to significantly increase cellular uptake and signal intensity [21,22]. The choice of fluorophore may be due to various reasons: (1) subsequent phage modification with an oncolytic drug or another molecule that has its own fluorescence; (2) the need for certain free amino acid residues; or (3) the subsequent use of the developed drug in an in vitro, in vivo, or ex vivo system. The choice of payload (the oncolytic drugs or other molecules) that can be conjugated to viral particles may be due to the following reasons: (1) increased binding to target cells and (2) effects on tumor cells. 

A great deal of research has gone into phage modification using conjugation chemistry or bioengineering, with the aim of elaborating upon theranostics or imaging agents. Breakthroughs in the history of phage-mediated gene therapy was performed by Larocca et al. They demonstrated that a filamentous phage modified by fibroblast growth factor (FGF2) and a functionally active green fluorescent protein (GFP) can target and deliver into mammalian cells [23]. Further, modified phages have also been shown to have applications for the characterization of breast cancer cells using fluorescence microscopy [24], for the intracellular delivery of functional exogenous proteins to a human prostate cancer cell line [25,26,27], for targeting and labeling abnormal collagens [28], and for using of phage particles as gene transfer vehicle [29]. The M13 bacteriophages can be genetically modified to express a RGD4C peptide on its shell to target cancer cells [30]. They can be vehicles for curative nucleic acids, and the decoration of their capsids with drugs and imaging dyes transforms the phages into theranostic platforms [18,26,31,32,33,34]. For example, refactored M13 bacteriophage for targeted imaging and drug delivery to prostate cancer cells in vitro has been developed: doxorubicin was attached to the M13 bacteriophage major coat protein pVIII, and the minor coat protein pIII displayed a peptide with affinity for SPARC (secreted protein, acidic, and rich in cysteine). The minor coat protein pIX was modified for the N-terminal display of a biotin acceptor peptide (BAP) [26]. Moreover, these chemically modified phages could be useful for scaffolds in a variety of detection schemes: lateral flow assays, tests for circulating tumor cells, surface-enhanced Raman spectroscopy, and as optical indicators or biosensors [35].

Phages themselves are considered safe for humans [36]. However, phage lysates can contain endotoxins of Gram-negative bacteria and protein toxins produced by many pathogenic bacterial species. A comparative analysis of cleaning from these impurities showed that PEG precipitation reduced the endotoxin-to-phage ratio of fHoEco02 by ∼20 fold [37]. This is much more effective than ultrafiltration or sucrose gradient ultracentrifugation. In any case, a small concentration of endotoxin remains. On the other hand, it was demonstrated that Ff phages associate with LPS and that LPS contributes to their anti-tumorigenic activity [38]. 

The goals of this research were not only to modify the filamentous phage by means of FAM-NHS ester for potential diagnostic or therapeutics applications but to develop a clear scheme for calculating the modification coefficient as well as to quantify the ability of the modified bacteriophage to penetrate into tumor cells compared to unmodified ones. 

In this work we used the chemical modification of the filamentous phage, which was based on a nonspecific modification of the carboxylic acid groups of aspartic acid or glutamic acid residues, the phenol group of tyrosine residues, or the free amino groups of the phage coat proteins, as they are the most reactive [9,10,11,12,13,14]. We optimized the M13 bacteriophage modification method with the FAM-NHS ester. It was shown that the concentration of both DMSO and the FAM fluorophore, which are present in the solution during modification, affects the titer of bacteriophages. Further, using the formula from the Beer–Lambert Law, we estimated the modification coefficients of the phages. Thus, it was obtained that the concentration of 4 mM FAM-NHS dye corresponded to the maximum of the modification coefficient of the phages. Next, it was necessary to check the binding efficiency of the modified tumor-targeting bacteriophages to tumor cells MDA-MB 231. Flow cytometry showed that bacteriophages modified with FAM-NHS dye with a concentration of more than 4 mM stained about 98% of the MDA-MB 231 cell population. In turn, confocal microscopy showed that bacteriophages modified with 4 mM FAM-NHS penetrated cells; phages were found in the immediate vicinity of the nucleus and in the cytoplasm and in the vesicles containing these bacteriophages. To obtain a therapeutic effect, it is necessary to estimate the number of phages that have penetrated into one tumor cell. We assumed that there was a certain optimal ratio for the number of phages to the number of cells, which could be quantified and used in the future to select the dose of a targeted drug that is currently under development. It was shown that with an increase in the concentration of phages with which MDA-MB 231 tumor cells are incubated, the number of phages that is able to penetrate into them increases. We also estimated the number of modified phages that penetrated into the tumor cells—it was significantly lower than expected and amounted to 96 phages per tumor cell. Perhaps this is due to the non-specific endocytic mechanisms of internalization of substances into the cell from the extracellular matrix besides the uptake via receptor-mediated endocytosis; in any case, further research is required to explain this mechanism [39]. These estimates are necessary for the further development of theranostic drugs based on tumor-targeting bacteriophages. Thus, tumor-targeting phages are promising molecules for the development of theranostics or imaging agents based on them. For the further use of bacteriophages, it is necessary to conduct full-fledged clinical trials in order to clarify issues related to dosage, methods of administration, and the possible therapeutic use of bacteriophage products [40]. 

## 4. Materials and Methods

### 4.1. Bacteria and Phages

*Escherichia coli* strain ER2738 with the genotype F’proA+B+lacIq Δ(lacZ)M15 zzf::Tn10(TetR)/ fhuA2 glnV Δ(lac-proAB) thi-1 Δ(hsdS-mcrB)5 was used as the host strain. Tumor-targeting M13 bacteriophage displaying the peptide (YTYDPWLIFPAN, hereinafter M13) was previously selected in our laboratory for the MDA-MB-231 cell line from the Ph.D.™-12 Phage Display Peptide Library Kit (New England Biolabs, United States) [41]. It was shown that this phage specifically binds with human breast adenocarcinoma MDA-MB-231 cells and penetrates into these cells through phagocytosis and clathrin-mediated endocytosis [42].

### 4.2. Cell Cultures

Experiments were performed with the epithelial human breast cancer cell line MDA-MB-231 (CDRI, Khimki, Russia). MDA-MB-231 cells were maintained in Leibovitz’s L-15 medium (Sigma, USA) supplemented with 10% of fetal bovine serum (Merck KGaA, Darmstadt, Germany), 1 mM L-glutamine (GlutaMAX, Thermo Fisher Scientific, Waltham, MA, USA), 100 units/mL of penicillin, 100 μg/mL of streptomycin, and 2.5 μg/mL of Fungizone (Antibiotic-Antimycotic, Thermo Fisher Scientific, Waltham, MA, USA). Cells were cultivated in an incubator at 37 °C in a saturated humid atmosphere.

### 4.3. Phage Clone Amplification

A suspension with a volume of 30 μL of tumor-targeting M13 bacteriophages (1 × 10^13^ PFU/mL) was incubated with *E. coli* in 25 mL of Luria Bertani (LB) medium supplemented with 10 μg/mL tetracycline for 4.5 h at 37 °C at 170 rpm. The bacterial cultures were centrifuged at 12,000 *g* for 10 min. The phages were precipitated with a 1/6 volume of supernatant of ice-cold PEG/NaCl (20% *w/v* PEG 8000/2.5 M NaCl) twice and were resuspended in 200 μL of phosphate buffer (PBS). Phage concentration was determined using the soft-agar overlay technique [42].

### 4.4. Sequencing of DNA

Phage DNA was isolated by denaturing coat proteins with iodide buffer (4 M NaI, 10 mM Tris-HCl, 1 mM EDTA) and was extracted with 96 % ethanol, according to the modified protocols [17]. The peptide-encoding DNA sequences were determined by Sanger sequencing at the Genomics Core Facility, Institute of Chemical Biology and Fundamental Medicine SB RAS, Novosibirsk. Sequencing was performed on an ABI 3130XL genetic Analyzer automatic sequencer (Thermo Fisher Scientific, Waltham, MA, USA) using the BigDye Terminator Cycle Sequencing Ready Reaction kit and sequencing primer “-96III” (5′-CCCTCATAGTTAGCGTAACG-3′). The nucleotide sequences were analyzed using the MEGA X software [43]. 

### 4.5. Modification of Phages

Fluorescein *n*-hydroxysuccinimide ester (FAM-NHS ester, Lumiprobe, Moscow, Russia) were prepared with dimethyl sulfoxide (DMSO). For phage modification, 100 μL of M13 bacteriophages (1 × 10^13^ PFU/mL) were incubated with 10 μL of 1M HEPES buffer (pH 8.5) and 20 μL of solution with varying concentrations of FAM-NHS (0.5 mM, 1 mM, 2 mM, 4 mM, 8 mM) at the total volume of 200 μL. The reaction was conducted for 20 h at 4 °C with stirring at 170 rpm. Following modification, the phages were precipitated once with 83 μL of PEG/NaCl (20% *w/v* PEG 8000/2.5 M NaCl) in 500 μL for 12 h at 4 °C with stirring at 170 rpm in the rotator. After incubation with PEG/NaCl, phages were centrifuged at 12,000 *g* at 4 °C for 20 min, and the supernatant was removed. The pellet was dissolved in 500 μL of PBS and was then centrifuged 2 min at 12,000 *g* at 4 °C and transferred to the new tubes. Next, we precipitated the phages with PEG/NaCl (1/6 volume) twice in 500 μL for 1 h on ice. The additional two rounds of PEG precipitation were needed to purify them phages from any residual free dye. The surface modification factor (dye per phage) of the virus was determined by UV-visible spectroscopy NanoDrop 2000c (Thermo Fisher Scientific, Waltham, MA, USA). 

### 4.6. Calculation of the Modification Coefficient of the Coat Protein of the M13 by the FAM Fluorophore

Here, we used Formula (1) obtained from The Beer–Lambert Law. The amount of FAM molecules attached to bacteriophage coat proteins pVIII was calculated based on the obtained absorption spectra of the modified samples at a wavelength of 494 nm. 

$\left(1\right)\;N_{FAM}=\frac{{OD}_{494}\;\times \;V\;\times \;N_{A}}{\varepsilon \;\times \;L},$ where *N_FAM_*—the amount of FAM molecules; *OD*_494_—the optical density of the solution at a wavelength of 494 nm (maximum of FAM absorption); *V*—the volume of solution (2 × 10^−6^ l); *ε*—molar extinction coefficient of FAM–NHS dye (75000 l mol^−1^ cm^−1^); L—optical path length (0.1 cm); and *N_A_*—Avogadro’s number.

At the moment, it is generally accepted to measure the number of bacteriophages in a solution using the soft-agar overlay technique, which allows an approximate estimate of the number of living bacteriophages. Within the framework of our study, it is necessary to apply more accurate methods; therefore, it was decided that the mass of the bacteriophages should be determined using spectrophotometry and that it should be divided by the mass of one bacteriophage, which was determined based on the physicochemical properties described by Newman J. et al. [20]. Based on the spectrophotometric data and according to Formula (2), the total number of phages was calculated.

$\left(2\right)\;N_{M13}=\frac{(OD_{269}\;-\;CF_{269}\;\times \;OD_{494})\;\times \;V}{\varepsilon \;\times \;L\;\times \;m_{1}},$ where *N*_*M*13_—the amount of M13 bacteriophages in the investigated volume; *OD*_269_—optical density of the solution at a wavelength of 269 nm (maximum absorption of M13 phages); *OD*_494_—the optical density of the solution at a wavelength of 494 nm; *V*—the volume of the solution (2 × 10^−6^ l); $CF_{269}$—correction factor adjusted for the amount of absorbance at 269 nm caused by the dye; *ε*—molar extinction coefficient of M13 bacteriophage (3.84 cm^2^/mg) [20,44]; *L*—optical path length (0.1 cm); and *m*_1_—the mass of one M13 bacteriophage (1.64 × 10^7^ Da) [20,45]. 

The modification coefficient of the M13 bacteriophage coat protein by the FAM fluorophore, depending on the dye concentration, was determined according to Formula (3). 

$\left(3\right)\;N=\frac{N_{FAM}}{N_{M13}}$, where *n*—modification coefficient; *N_FAM_*—the amount of FAM molecules; and *N*_*M*13_—the amount of M13 bacteriophages in the investigated volume. 

### 4.7. Immunocytochemistry

The previously described protocol was used with minor modifications [42]. MDA-MB-231 cells were seeded on glass slides (BD Falcon culture slides, Merck KGaA, Darmstadt, Germany) to a 70–80% monolayer and were incubated with 100 μM 2 × 10^10^ PFU/mL of phages for 2 h at 37 °C. Following incubation, the cells were washed with buffer (100 mM glycine, 0.5 M NaCl, pH 2.5) to remove any unbound phages, and they then fixed with 4% cold formaldehyde for 10 min and washed with PBS. Next, we incubated fixed cells with 0.2% Triton X100 for 10 min, and we then washed them twice with PBS. The bacteriophages were incubated with mouse Anti-M13 Bacteriophage Coat Protein g8p antibody (RL-ph2) (Abcam, 9225) dissolved in 1% BSA/PBS buffer for 45 min at 4 °C and were washed four times with 1% BSA/PBS buffer. Next, the cells were incubated with Goat Anti-Mouse IgG H&L (Alexa Fluor^®^ 647) (Abcam, 150115). The control cells were stained with 2.5 μM CellTracker ™ green CMFDA dye at 37 °C for 30 min, RT. All cells were stained with DAPI for 5 min in the dark and were washed with PBS. The slides were mounted with Prolong Diamond Mountant (Thermo Fisher Scientific, Waltham, MA, USA) and were coverslipped for viewing on a Zeiss LSM 710 AxioObserver confocal microscope (Zeiss AG, Jena, Germany).

### 4.8. Flow Cytometry

MDA-MB-231 cells were cultivated on 6-well cell culture plates (Merck KGaA, Darmstadt, Germany) until 80–90 % confluence was reached. The cells were detached with TripLE Express (Thermo Fisher Scientific, Waltham, MA, USA) for 3 min at 37 °C and were washed with PBS. Next, the cells were incubated for 2 h at 37 °C in 1.5 mL Eppendorf tubes with Leibovitz’s L-15 medium supplemented with 10% of FBS, 1 mM L-glutamine, 100 units/mL of penicillin, 100 μg/mL of streptomycin, and 2.5 μg/mL of fungizone and were washed with 1 % FBS. An amount of 200 μL of modified bacteriophages (1 × 10^8^ FPU/mL) in PBS-BSA Ca/Mg buffer (0.1% BSA, 1 mM CaCl_2_, 10 mM MgCl_2_ × 6H_2_O) were incubated with tumor cells on ice for 30 min. Following incubation, the cells were washed twice with 1 mL of 1% FBS. Further, the cells were counted using an automated cell counter «Countess» (Thermo Fisher Scientific, Waltham, MA, USA), and half of them used for further analysis on a BD FACS Canto II flow cytometer (Becton Dickinson, Holdrege, NE, USA).

### 4.9. Lysates Analysis

An amount of 200 μL of distilled water was added to the remaining part of the cells for cell lysis. Cells were lysed for 20 min at RT and were then centrifuged at 12,000 *g* for 10 min. After centrifugation, the supernatant was used for the evaluation of the phage concentration using the soft-agar overlay technique [42].

### 4.10. Statistical Analyses

Outcome variables are expressed as means ± standard deviations (SDs). Each experiment was repeated at least three times. Statistical analysis was performed using GraphPad Prism 6.01 (GraphPad Software, San Diego, CA, USA). Differences were considered to be significant if the *p*-value was < 0.05.

## 5. Conclusions

Initially, phage therapy was used for infectious diseases as an alternative to antibiotics, but research in this area is now rapidly developing and can bring us closer to creating “magic bullets” [46,47]. Now, phage modification can contribute to the development of the new molecular imaging probes and techniques for the early detection of cancer; moreover, bacteriophages themselves are particles with therapeutic potential. To develop a potential diagnostic or theranostic drug, we propose the use of tumor-targeting bacteriophages as carriers that have been selected using the phage display technique and that have been chemically modified with a fluorophore. The design of such targeting agents provides a number of advantages over other similar vehicles: the selection of specific bacteriophages for any cell culture through phage display; selective penetration into tumor cells; and the ability to monitor the penetration of the bacteriophages into the cell, which is possible with the use of a fluorescent dye. In the future, such modified phages could be conjugated with oncolytic agents as a therapeutic payload to develop theranostic drugs.

## Figures and Tables

**Figure 1 molecules-26-06564-f001:**
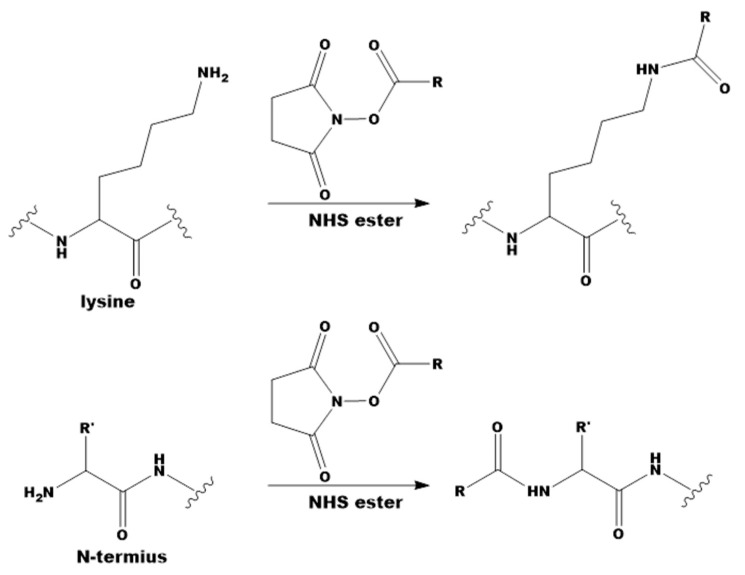
Free amine modifications [16].

**Figure 2 molecules-26-06564-f002:**
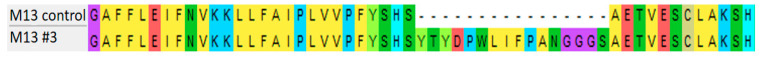
Alignment of the amino acid sequence containing the tumor-targeting peptide to the sequence of the coat protein of the M13 pIII phage containing no insert.

**Figure 3 molecules-26-06564-f003:**
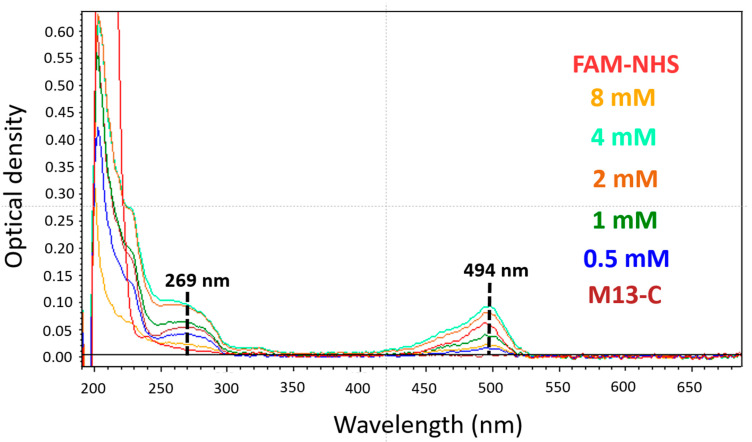
Absorption spectra of M13 bacteriophages modified at different concentrations of FAM-NHS dye. Ocher line-M13-FAM modified by 8 mM FAM-NHS in the reaction mixture, etc.: light green line-M13 modified by 4 mM FAM-NHS, brown line-2 Mm FAM-NHS, dark green line-1 mM FAM-NHS, blue line-0.5 mM FAM-NHS, burgundy line-M13-C (control).

**Figure 4 molecules-26-06564-f004:**
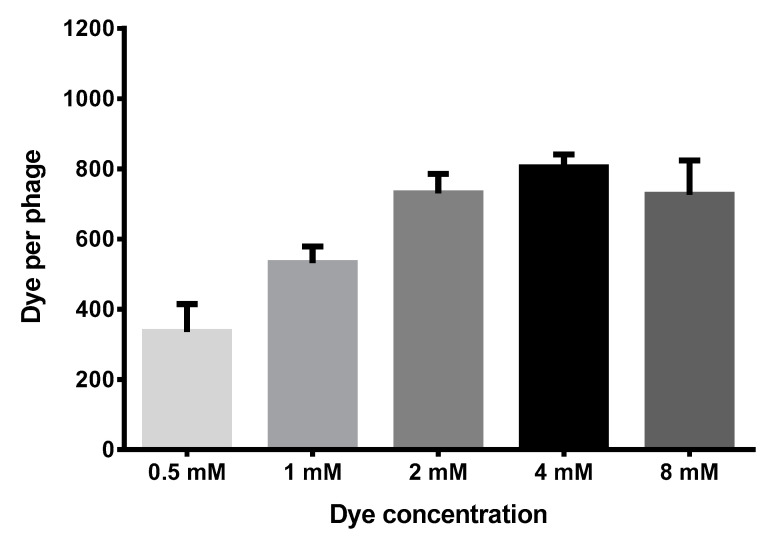
The dependence of FAM-NHS dye concentration on the modification coefficient. Note: FAM/phage is the modification coefficient of the bacteriophage coat protein by fluorescent dye.

**Figure 5 molecules-26-06564-f005:**
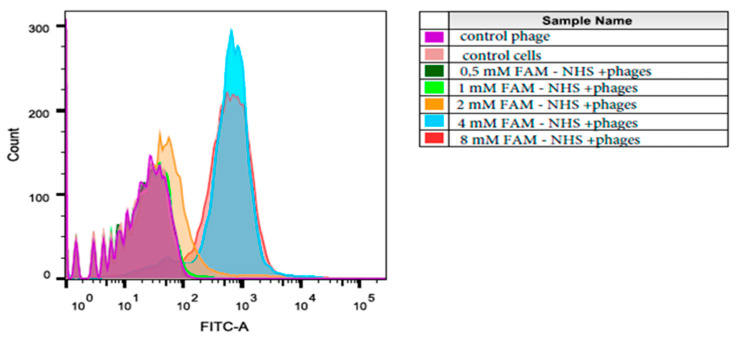
The flow cytometry of the tumor cell MDA-MB 231, which was incubated with M13-FAM phages. Notes: purple—control phage cells incubated with unmodified phages; pink—control cells without phages; dark green—cells incubated with phages modified by 0.5 mM FAM-NHS; light green—by 1 mM FAM-NHS; orange—by 2 mM FAM-NHS; blue—by 4 mM FAM-NHS; red—by 8 mM FAM-NHS.

**Figure 6 molecules-26-06564-f006:**
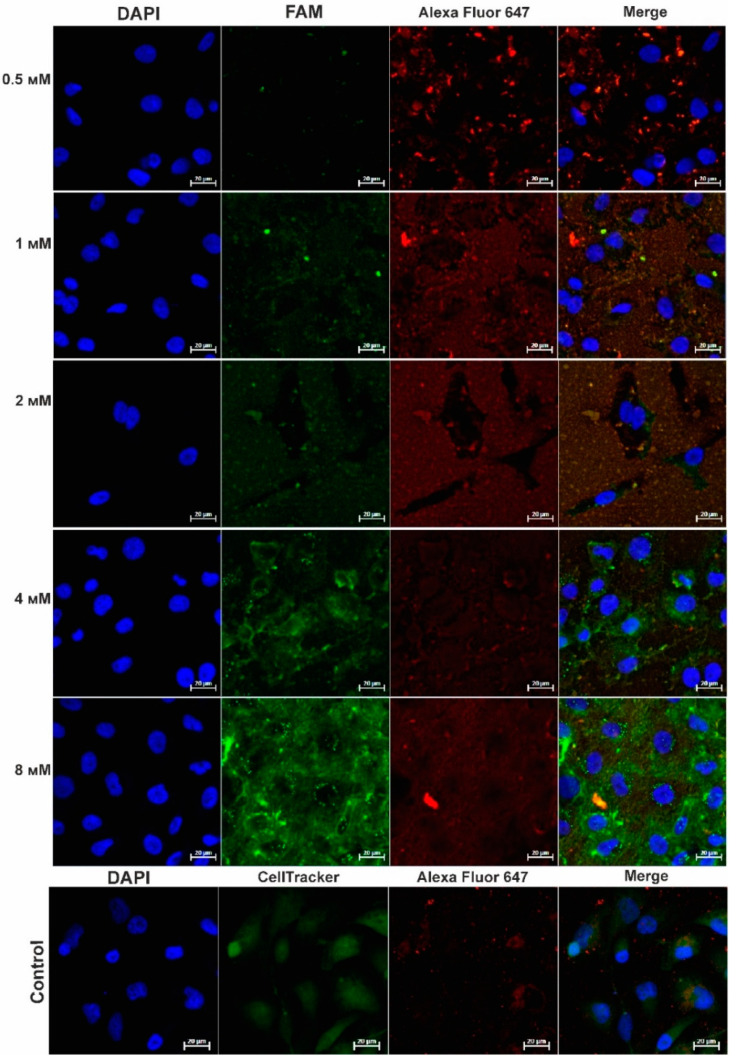
Penetration of bacteriophages M13-FAM and M13 (control) into MDA-MB-231 cells. Confocal microscopy was performed using the system: primary antibodies to the pVIII protein of the M13 bacteriophage plus secondary antibodies to mouse IgG (H + L) conjugated to Alexa Fluor 647 (red channel, 633 nm). For visualization, nuclei were stained with DAPI (blue channel, 405 nm). M13-FAM bacteriophage corresponds to the green channel, 488 nm. Controls were stained with CellTracker™ Green CMFDA (green channel, 488 nm). Lens 63×/1.40 Oil DIC M27.

**Figure 7 molecules-26-06564-f007:**
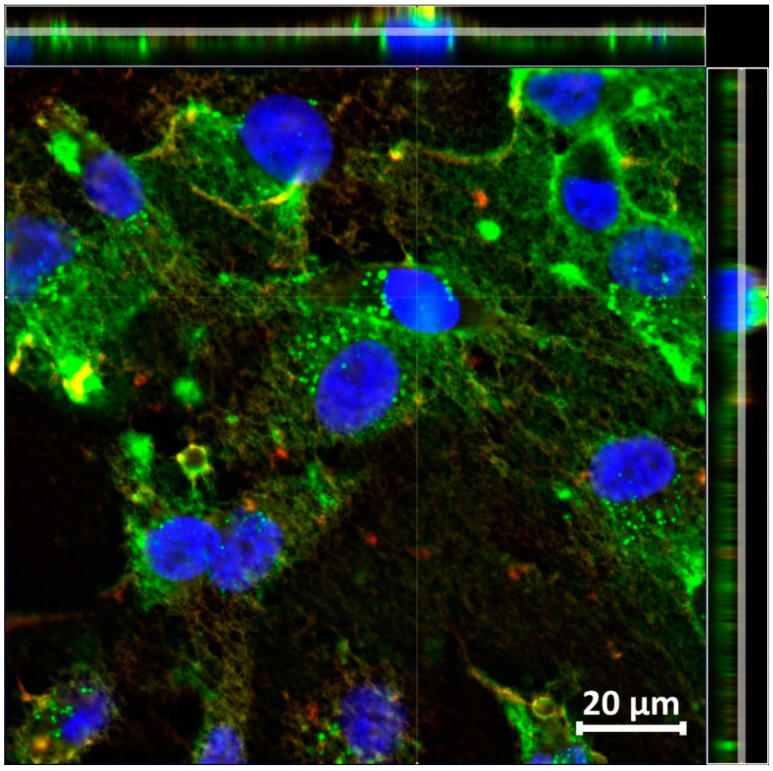
A 3D image of MDA-MB-231 cells incubated with modified M13-FAM bacteriophages (4 mM), Ortho image. Top-right—Z-axis images. Confocal microscopy was performed using the system: primary antibodies to the pVIII protein of the M13 phage and secondary antibodies to mouse IgG (H + L) conjugated to Alexa Fluor 647 (red channel, 633 nm). For visualization, nuclei were stained with DAPI (blue channel, 405 nm). The green channel, 488 nm, corresponds to the modified M13-FAM bacteriophage. Lens 63×/1.40 Oil DIC M27.

**Figure 8 molecules-26-06564-f008:**
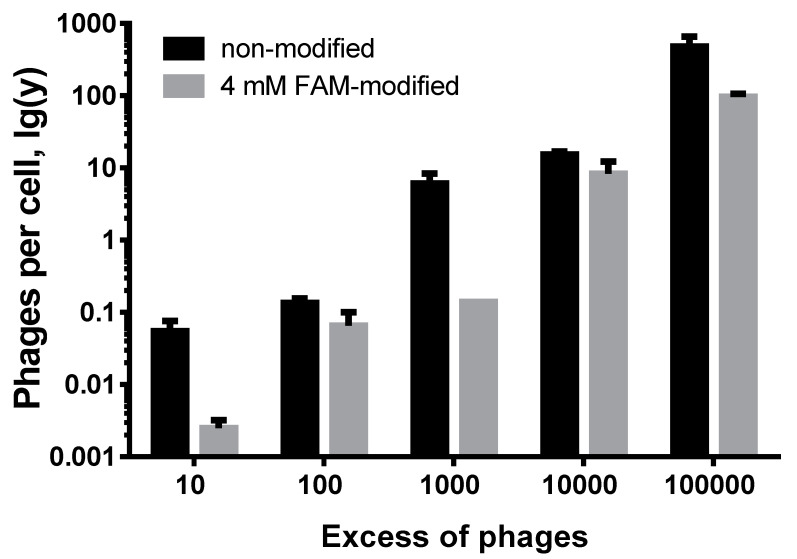
The dependence of the excess tumor-targeting phages incubated with fixed amount of tumor cells MDA-MB 231.

**Table 1 molecules-26-06564-t001:** The titers of modified and control phages M13 №3.

Concentration of FAM, mM	Titer, PFU/mL
0.5	8.3 × 10^10^
1	5.7 × 10^10^
2	3.3 × 10^10^
4	1.2 × 10^9^
8	5.0 × 10^8^
M13 (control) + DMSO	6.7 × 10^11^
M13 (control)	6.7 × 10^12^
M13 (initial)	1.0 × 10^13^

## Data Availability

Not applicable.

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
