# Peer review of "Modification of a Tumor-Targeting Bacteriophage for Potential Diagnostic Applications"

_molecules, 2021, doi:10.3390/molecules26216564_

Round 1

Reviewer 1 Report

The manuscript entitled "Modification of a tumor-targeting bacteriophage for potential diagnostic applications. " describes a well-justified method for using a modified bacteriophage M13 as a diagnostic tool. The manuscript is written with the excellent use of the English language, from authors who have a lot of experience in manipulating bacteriophage M13 under laboratory conditions.

In my opinion, the manuscript has some minor issues which should be corrected before considering the manuscript suitable for publication.

1) There are certain points in the results section which are not results but materials and methods. Please major revise both sections and move this information to materials and methods. I.e. lines 119-144, 70-77. This creates a lot of confusion and is a problem for the readers to evaluate the results presented. 

2) FAM-NHS are two abbreviations widely used. Nevertheless, due to the nature of the article and the broad readers that the article is going to attract, authors need to provide some more information and add some background for everyone in the introduction section. I understand that the authors choose to describe in detail the reaction with NHS in the discussion section, but I would advise some information to be supplemented in the introduction as well.

3) Please revise figure 5 in terms of quality. I understand what the authors want to point out but the figure is lacking the quality to detect differences between all the concentrations of FAM. 0.5 concentration is not appropriately displayed on the left side. 

4) Regarding the discussion section I believe that figure 8 belongs to the introduction among the information framing it. 

5) Using high titers of phages for sure means that authors inoculated also a high titer of endotoxins derived from the dead bacteria. It is possible that bacterial toxins can be a negative factor for using phages as "magic bullets" in tumors. I believe that authors could discuss that issue in one or two sentences.

Line 116 one with should be deleted.

Author Response

Response to Reviewer 1 Comments

The authors are grateful to reviewers for insightful comments on the manuscript and for the expressed questions and criticism, it helped to improve our manuscript a lot. Please find below our response to reviewer’s comments.

1) There are certain points in the results section which are not results but materials and methods. Please major revise both sections and move this information to materials and methods. I.e. lines 119-144, 70-77. This creates a lot of confusion and is a problem for the readers to evaluate the results presented.

Response: Corrected. We moved this information to the Section “Materials and Methods”.

2) FAM-NHS are two abbreviations widely used. Nevertheless, due to the nature of the article and the broad readers that the article is going to attract, authors need to provide some more information and add some background for everyone in the introduction section. I understand that the authors choose to describe in detail the reaction with NHS in the discussion section, but I would advise some information to be supplemented in the introduction as well.

Response:  Corrected. We have made changes in the Introduction section. We added there some more information about the reaction with NHS.

3) Please revise figure 5 in terms of quality. I understand what the authors want to point out but the figure is lacking the quality to detect differences between all the concentrations of FAM. 0.5 concentration is not appropriately displayed on the left side.

Response: Corrected. We have changed contrast of the image to show the differences more clearly. Unfortunately, it is impossible to improve the quality much more in the Word format. If it is necessary, we can present these pictures as supplementary files in jpeg format. Small details can be explored by using a full-size image.

Here, we consider that it is more important to show the differences between different concentrations.

4) Regarding the discussion section I believe that figure 8 belongs to the introduction among the information framing it.

Response: Corrected. As we have moved the text about the reaction from the discussion section to the introduction, we have moved the Fig. 8 also. All changes were highlighted in yellow.

5) Using high titers of phages for sure means that authors inoculated also a high titer of endotoxins derived from the dead bacteria. It is possible that bacterial toxins can be a negative factor for using phages as "magic bullets" in tumors. I believe that authors could discuss that issue in one or two sentences.

Response: Corrected. Thank You very much for this comment. We added the text in the discussion section: “Phages themselves are considered safe for humans [36]. But phage lysates can contain endotoxins of gram-negative bacteria and protein toxins produced by many pathogenic bacterial species. Comparative analysis of cleaning from these impurities showed that PEG precipitation reduced the endotoxin-to-phage ratio of fHoEco02 by ∼20 fold [37]. It is much better than ultrafiltration or sucrose gradient ultracentrifugation. In any case, a small concentration of endotoxin remains. On the other side it was demonstrated that Ff phages associate with LPS and that LPS contributes to their anti-tumorigenic activity [38].”

Line 116 one with should be deleted.

Response. Corrected.

We highlighted (yellow highlighted text) all changes made when revising the manuscript to make it easier for the Editors and reviewers to give a prompt decision on the manuscript.

Yours faithfully, Dymova Maya.

Reviewer 2 Report

Author reports the modification optimization of phages to MDA-MB 231 cells and the efficiency of phage particles penetration into MDA-MB 231 cells. FAM-NHS dye has been used to modify the phages, while immunofluorescence staining and flow cytometry have been employed to investigate the effectiveness of penetration.

In Fig. 2, the author used the optical densities (OD) at wavelength of 494 and 269 nm for the calculation of the modification coefficient (N). Although background correction was performed for OD at 494 nm, the OD at 269 was not corrected due to the contribution from other absorption peaks of M13. The tail of absorption peak at 230 nm can influence the OD at 269 nm, and thus, affecting the final value of N.

Based on the above rationale, it could be hard to conclude for the degree of phage modification for Fig. 3. Moreover, the error bars also indicate the possibility that 4 mM might not be the optimal modification, which could also influence the results of Fig. 4 and Fig. 5. More concentrations (one or two) of FAM-NHS dye could be performed and similarly for the other measurement (Fig. 4 and 5).

Please spell out the parameters of the modified phages in Fig. 7, that is the concentration of FAM (4 mM). Additionally, different FAM concentrations could also be shown (could be just for one ratio of excess phages) to demonstrate that the optimal FAM concentration has an increased/decreased affinity to the tumor cells.

For the conclusion section, please check the layout, the paragraph has been broken into two halves at the word “bacteriophages”.

Author Response

Response to Reviewer 2 Comments

The authors are grateful to reviewers for insightful comments on the manuscript and for the expressed questions and criticism, it helped to improve our manuscript a lot. Please find below our response to reviewer’s comments.

1). In Fig. 2, the author used the optical densities (OD) at wavelength of 494 and 269 nm for the calculation of the modification coefficient (N). Although background correction was performed for OD at 494 nm, the OD at 269 was not corrected due to the contribution from other absorption peaks of M13. The tail of absorption peak at 230 nm can influence the OD at 269 nm, and thus, affecting the final value of N.

Response: Indeed, we calculate the loading of FAM fluorophore on M13 bacteriophage only using the OD at 269 with the background correction performed for OD at 494 nm. These evaluations are detailed in several manuscripts (DOI: 10.1039/d0ra04086j, doi.org/10.1038/s41598-020-75205-3) and protocols (Morag O., Sgourakis N.G., Abramov G., Goldbourt A. (2018) Filamentous Bacteriophage Viruses: Preparation, Magic-Angle Spinning Solid-State NMR Experiments, and Structure Determination. In: Ghose R. (eds) Protein NMR. Methods in Molecular Biology, vol 1688. Humana Press, New York, NY. https://doi.org/10.1007/978-1-4939-7386-6_4). Therefore, we did not take into account the peak at 230 nm.

2). Based on the above rationale, it could be hard to conclude for the degree of phage modification for Fig. 3. Moreover, the error bars also indicate the possibility that 4 mM might not be the optimal modification, which could also influence the results of Fig. 4 and Fig. 5. More concentrations (one or two) of FAM-NHS dye could be performed and similarly for the other measurement (Fig. 4 and 5).

Response: You are absolutely right, the standard deviation does not indicate a statistical difference between 4mM and 8MM concentrations, but only a trend, a slight decrease. However, we see the same picture in the article on modifying the phage with another dye (doi.org/10.1021/bc900405q). There, too, with an increase in the concentration of the modifying agent (TMR-NHS), a similar picture is observed. The authors of the article explain this phenomenon - “The fluorescence intensity of M13-TMR was reduced because of the self-quenching that might have been caused by the close distance between rhodamine moieties, which could form nonfluorescent dimers”. Also, flow cytometry data showed that 4 mM modifying agent is optimal.

3). Please spell out the parameters of the modified phages in Fig. 7, that is the concentration of FAM (4 mM). Additionally, different FAM concentrations could also be shown (could be just for one ratio of excess phages) to demonstrate that the optimal FAM concentration has an increased/decreased affinity to the tumor cells.

Response: Corrected. We have changed the legend of fig. 7 (in the newest version it is fig. 8) and also added the information about concentration to the text. For this experiment, we used only 4 mM modified phages as the most promising for further studies based on the results of flow cytometry and confocal microscopy.

4). For the conclusion section, please check the layout, the paragraph has been broken into two halves at the word “bacteriophages”.

Response: Corrected.

We highlighted (yellow highlighted text) all changes made when revising the manuscript to make it easier for the Editors and reviewers to give a prompt decision on the manuscript.

Yours faithfully, Dymova Maya.

Round 2

Reviewer 2 Report

Thank you for addressing my comments and revising the script accordingly.

This manuscript is a resubmission of an earlier submission. The following is a list of the peer review reports and author responses from that submission.

Round 1

Reviewer 1 Report

The article titled “The Modification of a Tumor -Targeting Bacteriophage to Develop the Theranostic Drug”, authored by Dymova et al is interesting.

Regarding the experimental set, the methodology designed by the authors are apt and the results justify the claims of the article.

There are few major points that are required to be updated before this manuscript can be accepted for publication:

  1. The use of term “Theranostic” in the title should be avoided as the authors have showed the imaging part (diagnostic), yet not the therapeutic part (which they deem to do in future). Therefore, the title should be redrafted.
  2. The manuscript should be thoroughly revised for language, typos (fig 7 caption, exsess should be excess; L93 1,5 times should be 1.5 times etc.), grammar etc. Many lines are not understandable, and the reader must assume oneself what the authors mean. For ex. Line 13, phages to malignant breast cancer cells MDA-MB 231 were obtained. This line should be rewritten as “phages that target malignant breast cancer cells MDA-MB 231 specifically were obtained”. There are numerous such instances to be pointed out. Thereby, a thorough English editing is prerequisite for publication.
  3. There are many articles that utilized M-13 for cancer theranostic application, like doi: 10.1021/nn301134z, doi.org/10.1021/bc500339k, doi: 10.1021/sb300052u, DOI: 10.1016/j.biomaterials.2014.07.044, DOI: 10.1089/hum.1998.9.16-2393, doi.org/10.1021/bc900405q, doi.org/10.1007/s12274-011-0104-2 etc. The conjugation chemistry used for phage modification is also reported already. This article seems to follow similar trend and no novelty is involved. Therefore, it is important that the authors describe the previous research in this field and how this work is significantly different and important for the current research trend. This description is important to prove the novelty and importance of their work.
  4. The authors have quoted Reference 17 (L74) for the following statement “Earlier, in our laboratory, using the phage display technique a filamentous bacteriophage M13, displaying the peptide (YTYDPWLIFPAN, hereinafter M13) was found. It was shown that this phage specifically binds with human breast adenocarcinoma MDA-MB-231 cells and penetrates into these cells by phagocytosis and clathrin-mediated endocytosis”. Yet, I do not see any information about this peptide in the referenced article. Please recheck the reference and quote the correct one.
  5. The author has given no information regarding the target receptor expressed by MDA-MB 231 cells for the displayed peptide. It is necessary to elucidate the target for the peptide displayed phage. A simple cell protein lysate-based receptor binding studies should be done to elucidate the role of active targeting claim. The authors can refer the method given in the following article: doi:10.1093/protein/gzs013
  6. In the figure 7, what does the two-color bar (black and gray) indicates? Please give the information in Figure. Also, please explain why the modified phages exhibit low cell internalization? What factor yield to difference in internalization pattern between the unmodified and modified phages.

Author Response

Response to Reviewer 1 Comments

We thank reviewer for a thorough revision of our manuscript. Please find below our response to reviewer’s comments.

Point 1:

Title: I would change the title to “Modification of a Tumor-Targeting Bacteriophage as a Potential Theranostic Drug”

Response: We changed the title of manuscript from “Modification of a Tumor-Targeting Bacteriophage as a Potential Theranostic Drug” to “Modification of a tumor-targeting bacteriophage for potential diagnostic applications”.

Point 2:

Lines 15-17: I would remove the sentence that simply lists the methods used without any context

Response: In this case, we only follow the instructions for authors (https://www.mdpi.com/journal/biomedicines/instructions), where it says that methods should be mentioned in the abstract. In addition, the abstract should contain a maximum of about 200 words, and if we describe each method used in the work, we will not meet the volume requirements.

Point 3:

Line 98 says 10^13 PFU/ml but Table 1 gives the number 6.7 * 10^12 PFU/ml for M13. Which one is correct?

Response: We have corrected this part to be unambiguous. New details are highlighted in red (lines 93 and 94).

Point 4:

Figure 2: The light green line shows highest signals at 269 and 494 nm. Shouldn’t it be the yellow (8 mM) line? The yellow line shows very weak signals. Please check line colors and the legend.

Response: Theoretically, this should be so, but in practice, it turns out that a high concentration of the dye (8 mM) too strongly affects the titer of bacteriophages, as You can see in Table 1. The legend is correct. The peak at 269 nm indicates a concentration of the bacteriophages themselves, and, the peak at 494 nm corresponding to concentration of FAM. And the ratio between these two peaks (as You can see at Formulas 1-3) is more significant in this situation than a comparison of the peak levels corresponding to different samples (0.5 – 8 mM FAM-NHS +phages).

Point 5:

Figure 7: The x-axis title should read “Excess of phages”.

Response: Corrected.

Point 6:

In the Discussion section, please elaborate with a few sentences on the phages’ envisioned use as theranostic drug, since the term “theranostic” is in the title. What are potential agents used for diagnostics, what are potential agents for therapeutics (such as oncolytic agents)? What are other approaches in this direction?

Response: We added the text in the discussion and Conclusion sections. And since we changed the title according to your comments, we decided to delve into the topic of phage modification, and not into the development of theranostic drugs, in general. In the latter case, one should speak of promising novel radiopharmaceuticals for radionuclide therapy (radiolabelled antibodies, such as 131I-omburtamab, 89Zr/177Lu-labeled antibody, 177Lu-lilotomab; neurotensin receptor ligand 111In/177Lu-3B-227) [1], also for the PET imaging (the fibroblast activation protein inhibitor (FAPI) 68Ga-FAPI-04, and the first therapeutic applications of 90Y-FAPI-04) [2].

[1]      Langbein T, Weber WA, Eiber M. Future of theranostics: An outlook on precision oncology in nuclear medicine. J Nucl Med 2019;60. https://doi.org/10.2967/jnumed.118.220566.

[2]      Giesel FL, Kratochwil C, Lindner T, Marschalek MM, Loktev A, Lehnert W, et al. 68 Ga-FAPI PET/CT: Biodistribution and preliminary dosimetry estimate of 2 DOTA-containing FAP-targeting agents in patients with various cancers. J Nucl Med 2019;60. https://doi.org/10.2967/jnumed.118.215913.

We added the following text in the discussion section:

“Many researches have gone into phage modification using conjugation chemistry or bioengineering with the aim of elaboration of theranostics or imaging agents. Break-through in history of phage-mediated gene therapy was performed by Larocca et al. They demonstrated that a filamentous phage modified by fibroblast growth factor (FGF2) and a functionally active green fluorescent protein (GFP) can target and deliver into mammalian cells [25]. Further, it was shown the utility of the modified phages for the characterization of breast cancer cells using fluorescence microscopy [26], for the intracellular delivery of functional exogenous proteins to a human prostate cancer cell line [27–29], for targeting and labeling abnormal collagens [30], for using of phage particles as gene transfer vehicle [31]. The M13 bacteriophages can be genetically modified to express a RGD4C peptide on its shell to target cancer cell [32]. They can be vehicles for curative nucleic acids, and dec-oration of their capsids with drugs and imaging dyes transforms phages into theranostic platforms [22,28,33–36]. For example, refactored M13 bacteriophage for targeted imaging and drug delivery to prostate cancer cells in vitro was developed: doxorubicin was at-tached to M13 bacteriophage major coat protein pVIII, the minor coat protein pIII dis-played a peptide with affinity for SPARC (Secreted Protein, Acidic and Rich in Cysteine), and the minor coat protein pIX was modified for N-terminal display of a biotin acceptor peptide (BAP) [28]. Moreover, the chemically modified phages could be useful for scaffolds in a variety of detection schemes: lateral flow assays, tests for circulating tumor cells, sur-face-enhanced Raman spectroscopy, as optical indicator or biosensor [37].

The goals of this research were not only to modify the filamentous phage by FAM-NHS ester for potential diagnostic or therapeutics applications, but to develop a clear scheme for calculating the modification coefficient, as well as to quantify the penetration to tumor cells of the modified bacteriophage comparing to unmodified ones”….

….“Thus, tumor-targeting phages are promising molecules for the development of theranostics or imaging agents on their basis. For the further use of bacteriophages, it is necessary to conduct full-fledged clinical trials in order to clarify issues related to dosages, methods of administration, and possible therapeutic use of bacteriophage products [41]”.

We added also the text in the Conclusion section:

“Initially, phage therapy was used for infectious diseases as an alternative to antibiot-ics, but research in this area is now rapidly developing, which can really bring us closer to creating "magic bullets" [43,44]. Now phage modification can contribute to development of the new molecular imaging probes, techniques for early detection of cancer, besides bacteriophages themselves are particles with therapeutic potential”….

We highlighted (red highlighted text) all changes made when revising the manuscript to make it easier for the Editors to give a prompt decision on manuscript.

Yours faithfully, Dymova Maya.

Reviewer 2 Report

In this manuscript, the authors studied the effect of chemical modification of a tumor cell targeting bacteriophage on the infectious properties. They show that infectivity decreases by the degree of modification but is retained at high levels. Moreover, they show that the fluorescent dye conjugated to the phage is internalized by a cancer cell line and present in the cytoplasm after infection. This study provides an important step en route to developing cancer cell-targeting bacteriophages to deliver diagnostic and therapeutic molecules. I support the publication of the manuscript after the following points have been addressed.

Title: I would change the title to “Modification of a Tumor-Targeting Bacteriophage as a Potential Theranostic Drug”

Lines 15-17: I would remove the sentence that simply lists the methods used without any context

Line 98 says 10^13 PFU/ml but Table 1 gives the number 6.7 * 10^12 PFU/ml for M13. Which one is correct?

Figure 2: The light green line shows highest signals at 269 and 494 nm. Shouldn’t it be the yellow (8 mM) line? The yellow line shows very weak signals. Please check line colors and the legend.

Figure 7: The x-axis title should read “Excess of phages”

In the Discussion section, please elaborate with a few sentences on the phages’ envisioned use as theranostic drug, since the term “theranostic” is in the title. What are potential agents used for diagnostics, what are potential agents for therapeutics (such as oncolytic agents)? What are other approaches in this direction?

Author Response

Response to Reviewer 2 Comments

We thank reviewer for a thorough revision of our manuscript. Please find below our response to reviewer’s comments.

 Point 1:

The use of term “Theranostic” in the title should be avoided as the authors have showed the imaging part (diagnostic), yet not the therapeutic part (which they deem to do in future). Therefore, the title should be redrafted.

Response: We changed the title of manuscript from “Modification of a Tumor-Targeting Bacteriophage as a Potential Theranostic Drug” to “Modification of a tumor-targeting bacteriophage for potential diagnostic applications”.

Point 2:

The manuscript should be thoroughly revised for language, typos (fig 7 caption, exsess should be excess; L93 1,5 times should be 1.5 times etc.), grammar etc. Many lines are not understandable, and the reader must assume oneself what the authors mean. For ex. Line 13, phages to malignant breast cancer cells MDA-MB 231 were obtained. This line should be rewritten as “phages that target malignant breast cancer cells MDA-MB 231 specifically were obtained”. There are numerous such instances to be pointed out. Thereby, a thorough English editing is prerequisite for publication.

Response: The text of the article was proofreading before submitting the Journal by native English speaker. We have now corrected the text, but if thorough editing in English is still required, we will need more time than 5 days.

Point 3:

There are many articles that utilized M-13 for cancer theranostic application, like doi: 10.1021/nn301134z, doi.org/10.1021/bc500339k, doi: 10.1021/sb300052u, DOI: 10.1016/j.biomaterials.2014.07.044, DOI: 10.1089/hum.1998.9.16-2393, doi.org/10.1021/bc900405q, doi.org/10.1007/s12274-011-0104-2 etc. The conjugation chemistry used for phage modification is also reported already. This article seems to follow similar trend and no novelty is involved. Therefore, it is important that the authors describe the previous research in this field and how this work is significantly different and important for the current research trend. This description is important to prove the novelty and importance of their work.

Response: We added these references into discussion section, and include the sentences which described the scientific novelty of our research: “Many researches have gone into phage modification using conjugation chemistry or bioengineering with the aim of elaboration of theranostics or imaging agents. Break-through in history of phage-mediated gene therapy was performed by Larocca et al. They demonstrated that a filamentous phage modified by fibroblast growth factor (FGF2) and a functionally active green fluorescent protein (GFP) can target and deliver into mammalian cells [25]. Further, it was shown the utility of the modified phages for the characterization of breast cancer cells using fluorescence microscopy [26], for the intracellular delivery of functional exogenous proteins to a human prostate cancer cell line [27–29], for targeting and labeling abnormal collagens [30], for using of phage particles as gene transfer vehicle [31]. The M13 bacteriophages can be genetically modified to express a RGD4C peptide on its shell to target cancer cell [32]. They can be vehicles for curative nucleic acids, and dec-oration of their capsids with drugs and imaging dyes transforms phages into theranostic platforms [22,28,33–36]. For example, refactored M13 bacteriophage for targeted imaging and drug delivery to prostate cancer cells in vitro was developed: doxorubicin was at-tached to M13 bacteriophage major coat protein pVIII, the minor coat protein pIII dis-played a peptide with affinity for SPARC (Secreted Protein, Acidic and Rich in Cysteine), and the minor coat protein pIX was modified for N-terminal display of a biotin acceptor peptide (BAP) [28]. Moreover, the chemically modified phages could be useful for scaffolds in a variety of detection schemes: lateral flow assays, tests for circulating tumor cells, sur-face-enhanced Raman spectroscopy, as optical indicator or biosensor [37].

The goals of this research were not only to modify the filamentous phage by FAM-NHS ester for potential diagnostic or therapeutics applications, but to develop a clear scheme for calculating the modification coefficient, as well as to quantify the penetration to tumor cells of the modified bacteriophage comparing to unmodified ones “.

Point 4:

The authors have quoted Reference 17 (L74) for the following statement “Earlier, in our laboratory, using the phage display technique a filamentous bacteriophage M13, displaying the peptide (YTYDPWLIFPAN, hereinafter M13) was found. It was shown that this phage specifically binds with human breast adenocarcinoma MDA-MB-231 cells and penetrates into these cells by phagocytosis and clathrin-mediated endocytosis”. Yet, I do not see any information about this peptide in the referenced article. Please recheck the reference and quote the correct one.

Response: We added the reference (https://doi.org/10.1371/journal.pone.0160980 ) in the beginning of the section “ 2.1. Obtaining modified bacteriophages M13-FAM”.

Point 5:

The author has given no information regarding the target receptor expressed by MDA-MB 231 cells for the displayed peptide. It is necessary to elucidate the target for the peptide displayed phage. A simple cell protein lysate-based receptor binding studies should be done to elucidate the role of active targeting claim. The authors can refer the method given in the following article: doi:10.1093/protein/gzs013

Response:

Elucidation of cellular targets for the peptide is a separate fundamental problem. In the context of our work, it is sufficient to understand that bacteriophages have selective penetration. The used bacteriophages presumably penetrate in cells via phagocytosis and clathrin-dependent endocytosis, it was shown using electron microscopy by our colleagues (https://doi.org/10.1371/journal.pone.0160980 Fig 3). We added the reference in the beginning of the section “ 2.1. Obtaining modified bacteriophages M13-FAM”.

Point 6:

In the figure 7, what does the two-color bar (black and gray) indicates? Please give the information in Figure. Also, please explain why the modified phages exhibit low cell internalization? What factor yield to difference in internalization pattern between the unmodified and modified phages.

Response: Figure 7 was corrected. We also added the sentence: “We assume that a tumor-targeting peptide can be modified with a fluorescent dye, which leads to a decrease in the affinity of the modified phages for tumor cells”.

Corrected text is displayed in red color.

We highlighted (red highlighted text) all changes made when revising the manuscript to make it easier for the Editors to give a prompt decision on manuscript.

Yours faithfully,

Dymova Maya.